# Multiple Case Studies in German Children with Dyslexia: Characterization of Phonological, Auditory, Visual, and Cerebellar Processing on the Group and Individual Levels

**DOI:** 10.3390/brainsci12101292

**Published:** 2022-09-25

**Authors:** Carolin Ligges, Thomas Lehmann

**Affiliations:** 1Department of Child and Adolescent Psychiatry, Psychosomatic Medicine and Psychotherapy, Jena University Hospital, Friedrich Schiller University Jena, 07743 Jena, Germany; 2Institute of Medical Statistics, Computer and Data Sciences, Jena University Hospital, Friedrich Schiller University Jena, 07743 Jena, Germany

**Keywords:** dyslexia, multiple deficit, cognitive deficit, behavioral data, shallow orthography, consistent orthography

## Abstract

Background: The underlying mechanisms of dyslexia are still debated. The question remains as to whether there is evidence of a predominant type of deficit or whether it is a multideficit disorder with individual profiles. The assumptions of which mechanism causes the disorder influences the selection of the training approach. Methods: A sample of German neurotypical reading children (NT) and children with dyslexia (DYSL) was investigated with a comprehensive behavioral test battery assessing phonological, auditory, visual, and cerebellar performance, thus addressing performance described in three major theories in dyslexia. Results: In the present sample using the test battery of the present study, DYSL had the strongest impairment in phonological and auditory processing, accompanied by individual processing deficits in cerebellar performance, but only a few in the investigated visual domains. Phonological awareness and auditory performance were the only significant predictors for reading ability. Conclusion: These findings point out that those reading difficulties were associated with phonological as well as auditory processing deficits in the present sample. Future research should investigate individual deficit profiles longitudinally, with studies starting before literacy acquisition at as many processing domains as possible. These individual deficit profiles should then be used to select appropriate interventions to promote reading and spelling.

## 1. Introduction

Individuals affected by dyslexia have substantial impairments in the acquisition of grapheme–phoneme correspondence, fluent and accurate reading, and spelling. Dyslexia is not caused by a general cognitive impairment or a lack of an opportunity to learn [1] and it affects about seven [2] to 18 [3] percent of the population, depending on diagnostic criteria [4,5,6]. Even today, individuals affected by dyslexia do not yet receive early and sufficient support, and thus dyslexia still has far-reaching consequences for their psychosocial, educational, and socio-economic development [7,8]. Therefore, it is inevitable to uncover the mechanisms of origin of the disorder and to derive adequate therapeutic measures from it.

There is a consensus in the scientific community that dyslexia is a neurobiological disorder with a strong genetic basis [9,10,11,12,13]. These mechanisms affect the acquisition and automation of the reading and spelling process. Nevertheless, the debate about the exact nature of the neurobiological and related cognitive deficits continues unabated [5,14,15,16,17]. In particular, the question as to whether a single core deficit can cause the variety of observed deficits, or whether the disorder should rather be viewed as a syndrome with possible deficits at different levels leading to the reading and spelling problems, remains open [15,18,19,20,21,22].

Within the plenitude of hypotheses that currently attempt to explain the origin of dyslexia, there are three major hypotheses explaining the development of the disorder via alternative pathways [23,24]. In order to acquire reading and spelling properly, some basic linguistic skills such as phonological awareness, phonological recoding in working memory, and phonological recall from long-term memory are of high relevance [25]. The “phonological deficit theory” explains dyslexia as being caused by deficiencies in those phonological processing abilities [25,26]. There is empirical evidence that phonological processing deficits can be observed in individuals with dyslexia already prior to the onset of formal education [13]. Against this background, the phonological deficit is often described as the core deficit in dyslexia [27,28].

However, many studies also show processing deficits in individuals with dyslexia in the sensory (visual and auditory) and motor systems that cannot be explained by the phonological deficit hypothesis [29,30,31,32,33,34,35].

Since individuals with dyslexia sometimes perform worse in the perception of fast-moving or low-contrast visual stimuli compared to typical reading and spelling individuals, these deficits are supposed to be indicative of an impairment of the magnocellular visual system underlying dyslexia [34,36,37]. The visual system consists of different neural elements, each of which is responsible for perceiving and processing different informational aspects of a visual stimulus [38]. Two of those elements are the parvocellular-system, which is relevant for the processing of high-contrast, slow-moving, color stimulus characteristic and the magnocellular system, which is responsible for the perception and processing of fast-moving, low-contrast stimulus characteristics as well as for the control of eye movements [39]. The eye movements controlled by the magnocellular system are supposed to play a major role in reading problems: a magnocellular deficit leads to unstable eye control, thereby to unstable visual perception, and finally to reading problems. Furthermore, unstable eye movements are not only supposed to impair visual word perception but also to impair grapheme–phoneme mapping, i.e., the transformation from the written letter to the speech sound, since the letters to be named cannot be sufficiently fixated [34]. 

In addition, the auditory sensory system is supposed to contain magnocells, which are supposed to be responsible for the perception of fast-changing acoustic information in the range of milliseconds. Successful (written) speech processing requires precise temporally high-resolution processing capacities, for example, for the perception of stop consonants, which contribute significantly to the disassembly of the auditory speech sound stream. There is empirical evidence for temporal auditory processing deficits in people with dyslexia. Against this background, the “auditory temporal processing deficit theory” [40] describes how this auditory deficit at a sensoric level impairs general speech perception and leads to reading and spelling problems.

The visual and auditory processing deficits are summarized in the “magnocellular deficit hypothesis” [34,41]. According to this theory, phonological deficits are the secondary consequence of this fundamental magnocellular deficit. These auditory and visual processing deficiencies are supposed to compromises the development of phonological awareness, which in consequence leads to reading difficulties.

The “cerebellar deficit theory” [42] is based on the observations that individuals with dyslexia perform worse on tasks that require some degree of automation of motor processes, which represent cerebellar functions. According to Nicolson & Fawcett, the problems in reading and spelling in the context of dyslexia are supposed to be the epiphenomenon of a more general learning disorder, namely, the impaired ability to achieve a sufficient level of automation of learned skills.

For an overview of the three deficit theories, please refer to Table 1 as well as to [43,44,45]. Building on Frith’s causal model of the phonological deficit in dyslexia [45], Ramus expands the framework to a neurological model of dyslexia by incorporating the explanatory approaches of the magnocellular and cerebellar deficit hypotheses [43,44]. Table 1 refers to those models by depicting the neurobiological, cognitive, and behavioral background of those deficit theories.

As elegant as these hypotheses may seem at first glance, they cannot explain all observable deficits comprehensively in dyslexia. In addition, there is the problem that previous empirical studies, which investigated only one of the various hypotheses at a time, have led to heterogeneous findings [23]. Consequently, it is still unclear as to whether the various deficits might suggest that dyslexia should be divided into subgroups depending on the focus of the deficit [46,47] or whether these deficits are symptoms of a superordinate mechanism. Due to the heterogeneity of the findings of those “single deficits,” successive studies were conducted, assessing various processing deficit theories in the same sample to investigate whether there is stronger evidence for one of the investigated disorder theories [23,47,48,49,50,51].

Despite the immense effort put into such studies (large test batteries were used, collection of behavioral and neurobiological data with various neuroscientific methods), drawing valid conclusions is hampered by the fact that the studies come from different language systems, use different test batteries, and look at different age groups. The different language systems are of particular importance here, since especially in languages that exhibit stringent grapheme–phoneme correspondence (“shallow” or “consistent” orthography), phonological processes play a particularly important role for reading and spelling acquisition [52].

The aim of the present study was thus to investigate multiple processing domains (phonology, auditory, visual, and cerebellar processing) in the same sample of neurotypical normal reading and dyslexic children. We wanted to identify whether in a shallow and consistent language system (German) there is evidence for single vs. multidimensional processing impairments in children with dyslexia, and which individual performance profiles prevail.

The study examined four research questions in order to examine performance profiles in neurotypical normal reading children (NT) and children with dyslexia (DYSL) at the group as well as the individual level: (1) Is there evidence on the group level that DYSL show significantly poorer performance than NT in one of the four processing domains (phonological, auditory, visual, or cerebellar domain)? (2) Do more DYSL show deviant performances on either the level of a sum-score of z-standardized performance data (“index”) or on the level of the single subtests used to assess the four processing domains? (3) Is there evidence for typical individual deficit profiles of DYSL? (4) Which of the investigated domains has the most predictive value for the overall reading performance in the whole sample?

## 2. Materials and Methods

For more details about the standardized test material, please refer to the Appendix A. For further details about the behavioral test battery tasks, please refer to the Appendix A.

### 2.1. Sample and Sample Criteria

We investigated 21 neurotypical normal reading children as well as 20 children with dyslexia matched according to age and nonverbal IQ. The nonverbal IQ of all children had to be ≥ 85. Sample criteria for DYSL were a double discrepancy (≥1.5 standard deviations (*SD*) between nonverbal IQ and reading and spelling performance as well as reading and spelling performance < percentile rank (PR) 15). Sample criteria for NT was discrepancy between nonverbal IQ and reading as well as spelling performance <1 *SD* as well as reading and spelling performance > PR 15. Sample criteria were assessed during a diagnostic session using several standardized tests.

Children with uncorrected impairment of sight or hearing, bilingual education, or neurological or psychiatric disorders (especially ADHD) were excluded from the study on the basis of the information of a detailed clinical interview as well as performance in an attention test (KiTAP, [53]). All children were right handed according to a standardized test for handedness (HDT, [54]). As a measure of nonverbal intelligence, the performance (nonverbal) IQ of HAWIK-III [55] was assessed. 

Reading fluency and accuracy both in single-word as well as in text reading was assessed by means of a standardized reading test (ZLT, [56]). In this test, children read different lists of single words as well as different texts respective to class level. The test acquires scores for the time needed as well as the errors made while reading the single words and texts. 

Spelling performance was assessed by dictation of a gap text by means of standardized spelling tests appropriate for the respective class level (WRT 6+ [57], DRT 4 [58], and DRT5 [59]). The respective test acquires the number of wrongly spelled words. For details of sample description, please refer to Table 2 and Table 3.

### 2.2. Behavioral Test Battery

After the diagnostic session in which the sample criteria were assessed, we conducted a second session in which we applied a behavioral test battery using tasks that already have been used in the literature to assess phonological, auditory, visual, and cerebellar performances. Due to the multitude of existing theories in dyslexia, this test battery does not cover all possible levels of performance and perception discussed within the range of dyslexia. On the basis of ethical considerations regarding children’s endurance limits in the context of research studies, we designed a test battery with a processing time of two hours that focused on a selection of test procedures commonly used in previous studies to operationalize these three processing domains. Please refer to the Appendix A for details of the behavioral test battery.

#### 2.2.1. Assessment of Phonological Performance

Three areas of phonological processing play a major role in learning to read and write: *phonological awareness*, *recoding in phonological working memory*, and *access to the semantic lexicon* (see [25] for a review). The following tests were used to assess these three phonological processing areas.

***Phonological awareness*** describes the ability to analyze and operate with the sound structure of the spoken language and is supposed to be a key factor for successful development of reading and spelling skills. Phonological awareness was tested using five subtests of the PTB [60] and BAKO [61,62]: (1) segmentation of pseudoword in its speech sounds, (2) vowel substitution, (3) swapping of phonemes, (4) judging similarity of speech sound categories, and (5) reverse speech sounds of a word. 

***Phonological recoding in the working memory*** describes the ability to keep the representation of acoustic sound information of written symbols “alive” in short-term memory and is very crucial for the beginning reader. The capacity of the phonological working memory was tested by the repetition of digit sequences using a subtest of the HAWIK-III. 

***Phonological recoding with access of the semantic lexicon*** is the ability of a (fast and automatized) recall of learned/knowledge-based elements within the semantic memory and is also an important ability needed for a fast and efficient reading process. This memory access was tested by the rapid naming of black and white as well as colored objects (BISC, [63]), as well as the rapid naming of lists of letters. 

***Pseudoword reading*** We additionally administered non-standardized lists of one- and three-syllable pseudowords in order to operationalize reading processes that can only be solved by grapheme–phoneme conversion processes [27].

#### 2.2.2. Assessment of Auditory Performance

We used three different auditory paradigms in order to investigate auditory sensory processing at different levels, from basic sensory auditory processing of single tones (***Pitch differentiation)*** and ***Tone Sequences*** without linguistic reference or content to auditory ***Syllable*** processing which involves processing of auditory and linguistic information. ***Pitch differentiation*** operationalized a control task of auditory processing on the simplest level without the need of the processing of auditory spectral information.

***Tone Sequenc****e* addresses auditory processing of tones on a slightly more complex level than in ***Pitch Differentiation.*** The ***Tone* *Sequences*** operationalized processing of rapidly changing acoustic information without linguistic content [64].

The task ***Syllables*** (differentiation of two acoustically presented syllables) was used to operationalize auditory processing of auditory information with rapid changes and linguistic content [65].

#### 2.2.3. Assessment of Visual Performance

We used several tasks with different visual stimulus material to address the magnocellular and parvocellular processing pathways of the visual system.

***Pattern Sequence*** (differentiation of two visually presented, non-moving, high-contrast stimuli operationalized visual processing addressing the visual *parvocellular* system [66]). ***Coherent Motion*** (detecting coherent motion onset in a cloud of randomly moving dots) addressed the visual *magnocellular* system [67]. ***Coherent Color*** (detection of coherent color change within a cloud of randomly moving dots) addressed the visual *parvocellular* system [67]. ***Stationary*** (non-moving) and ***moving*** sine wave vertical and horizontal gratings were applied in order to evoke activation in different parts of the visual system. Stationary visual stimuli should evoke activity essentially in the parvocellular system, and moving visual stimuli should evoke activity in the magnocellular system [68].

***Mental rotation tasks of visual stimuli*** (letters and objects) was used as a visual control task [69].

#### 2.2.4. Assessment of Cerebellar Performance

We used two tasks from the test battery NEPSY [70] in order to operationalize cerebellar performance in the form of sensorimotor processing. The subtest ***Fingertapping*** assesses the child’s finger dexterity, motor speed, and rapid motor programming.

The subtest ***Statue*** assesses motor persistence and inhibition.

### 2.3. Statistical Analysis

In the group of neurotypical readers, only one variable was missing for one person. In the group of dyslexic readers, five persons had missing values, yet those data were missing completely at random. We used the multiple imputation feature integrated in SPSS (version 28.0) in order to be able to keep all children in the analysis. Multiple imputation of missing values was performed separately for each group. We chose fully conditional specification (FCS) as the imputation method with five imputations for the missing data.

We conducted four different analysis approaches to examine performance at the group level, at the subtest level, and on the level of the performance profile of a single individual.

Firstly, we investigated the question as to whether DYSL show worse performances than NT in phonological, auditory, visual, and cerebellar variables on the group level. We compared the different parameters assessed in the behavioral test battery by means of two-sided independent samples *t*-tests between NT and DYSL. We applied Bonferroni correction to account for the problem of multiple comparisons [71]. The significance level was set at α = 0.05.

Secondly, we addressed the question as to whether there is evidence that more DYSL show deficient performances in a particular processing domain. In order to investigate this question, we computed so called “indices.” We combined z-standardized variables into an index of a specific performance area. The subtests included in the index of the respective domain are depicted in Table 4 for phonological performance, Table 5 for auditory performance, Table 6 for visual performance, and Table 7 for cerebellar performance.

For the indices, we standardized the variables using the mean and standard deviation of the neurotypical normal reading group, using this group as a “norm population”. In this way, it was possible to describe whether performance of the DYSL differed from the values of the norm population and which DYSL individuals did so. To establish comparability previous studies [23,48], we used two thresholds (1 and 1.65 *SD* from the sample mean) in order to define a particular performance as “deficient.” In order to analyze whether the percentage of individuals with deviations differed significantly between the groups, we applied the two-sided Fisher’s exact test.

Accordingly, we looked thirdly at individual profiles by looking at all standardized variables of all subtests that made up the respective indices. Several multicausal studies depicted individual performance profiles as intersection diagrams in order to demonstrate interindividual variance of performance profiles [23,48,51]. In the present study, we introduced “heatmaps” in order to examine and represent individual performance profiles (see Figures 6–10 in Section 3.3.3). Separate heatmaps were created for the indices and for each processing domain. The columns on the right half of the heatmaps contain the individuals of the NT, and the columns on the left side of the heatmaps contain the individuals of the DYSL. One column corresponds to one individual in the respective group. The number of the individual is in the column header of the heatmaps. In the rows of the heatmaps, the subtests of a particular processing domain were listed. To represent deviant performance using the 1 *SD* threshold, the individual’s performance on the respective subtest was coded as “0” and “1.” Cells containing a “1” were additionally colored red. The “0” corresponded to normal performance of the individual in the respective subtest, and the “1” corresponded to a deficient performance in this subtest. This coding number and especially the color-coding were intended to reveal whether typical individual deficit profiles in a specific processing domain could be observed.

Fourthly, we performed regression analyses in order to investigate which variables had the most predictive value for reading performance. We performed multiple linear regression analysis over the whole sample (NT and DYSL together) with the dependent variable “reading” and the predictors phonological awareness, phonological working memory, rapid automatized naming (RAN) performance, auditory reaction time (rct), auditory errors, visual magnocellular total performance, visual parvocellular total performance, and cerebellar total performance.

For the consideration of the results of the different performance ranges and test results, please note that most parameters are either error rates or reaction times. Higher values hereby represent worse performance. This is different for the phonological awareness tests, the test “digit repetition” (phonological working memory), and the test “statue” (cerebellar assessment): these variables represent test scores. In these scores, higher values represent better performance. Since we also computed indices (mean scores of z-standardized variables), we inverted the values of phonological awareness tests, digit repetition, and statue for better comparability. This means that in all indices, higher values represent worse performance. Moreover, note that for the t-tests, all levels of significance have been Bonferroni corrected according to the number of comparisons calculated for the respective analysis area.

## 3. Results

### 3.1. Sample Criteria and Attention Performance

Table 2 demonstrates that the sample of NT and DYSL was well matched according to age and nonverbal IQ (NT: mean 11.48 years (*SD*: 0.75); DYSL: mean 11.38 years (*SD*: 1.08)) and nonverbal IQ (NT: mean IQ 115.86 (*SD*: 12.18); DYSL: mean IQ 114.40 (*SD*: 9.81)). As required by the sample criteria, reading and spelling performance of the NT was within the normal range, and the performance of the DYSL were below average. Additionally, discrepancy between reading and spelling was <1 *SD* for NT and >1.5 *SD* for DYSL. The handedness score in both groups indicated clear right-handedness in the study participants. Attention performance (see Table 3) did not show any significant differences between the groups, neither in reaction time nor the different error types (commission and omission).

### 3.2. Group-Level Analyses

#### 3.2.1. Analysis of Phonological Performance

DYSL showed highly significant lower performances than NT in all three phonological processing areas (see Table 4). These differences were observed in subtests of phonological awareness (sound categorization (DYSL: mean 6.90 (*SD*: 2.13), NT: mean 9.10 (*SD*: 1.04), *p* < 0.001), word reversal (DYSL: mean 3.53 (*SD*: 2.93), NT: mean 6.76 (*SD*: 2.45), *p* < 0.001), pseudoword reading (for example, 3 syl PW rct (DYSL: mean 55.20 (*SD*: 19.82), NT: mean 31.14 (*SD*: 11.21), *p* < 0.001)), phonological recoding in working memory (DYSL mean 9.20 (*SD*: 2.09), NT: mean 12.60 (*SD*: 3.63), *p* < 0.001), and one subtest of phonological recoding with access to the semantic lexicon (reaction time for b/w-objects (DYSL: mean 28.80 (*SD*: 7.87), NT: mean 22.33 (*SD*: 3.12), *p* < 0.017)).

#### 3.2.2. Analysis of Auditory Performance

Regarding the auditory processing with and without linguistic content, there were also some highly significant performance differences between NT and DYSL (see Table 5). For single tone processing, but not for tone sequences and syllables, DYSL reacted significantly faster than NT (DYSL: mean 481.09 (*SD*: 97.02), NT: mean 570.79 (*SD*: 72.38), *p* < 0.01). Yet, in all investigated auditory performances, DYSL made significantly more errors, and the difference of total error number was large. The largest difference was observed for the tone sequences (single tone error (DYSL: mean 16.35 (*SD*: 7.82), NT: mean 6.67 (*SD*: 5.97), *p* < 0.001), tone sequence error (DYSL: mean 46.60 (*SD*: 25.27), NT: mean 11.95 (*SD*: 15.23), *p* < 0.001), and syllable errors (DYSL: mean 9.20 (*SD*: 7.14), NT: mean 3.76 (*SD*: 4.76), *p* < 0.001). In order to search for hints as to why DYSL made so much more mistakes for tone sequence, we also looked at the performances regarding the different interstimulus intervals (ISI) of the tone sequences (see Appendix A).

Regardless of the ISI, DYSL were faster than NT for all ISIs, but this was not significant for any ISI. Total error rates differed between the groups for all ISIs, not only the short ones. (For example, TS_rct_25ms (DYSL: mean 1165.08 (*SD*: 454.08), NT: mean 1052.38 (*SD*: 347.00), *p* = 1.000), TS_err_25ms (DYSL: mean 0.95 (*SD*: 1.20), NT: mean 2.80 (*SD*: 1.33), *p* < 0.001).)

#### 3.2.3. Analysis of Visual Performance

We found no statistical differences between NT and DYSL in their performance in visual tasks addressing the visual magnocellular system. In contrast, we found significant group differences in a task assessing the visual parvocellular system. Here, the error rate for the pattern sequence task showed a significantly worse performance for DYSL compared to NT (PS error (DYSL: mean 2.00 (*SD*: 2.12), NT: mean 5.39 (3.83) *p* < 0.06; see Table 6).

#### 3.2.4. Analysis of Cerebellar Performance

After Bonferroni correction, there were no significant group differences for cerebellar performance, yet for tapping, there was a trend towards worse performance in DYSL (DYSL: mean 45.57 (*SD*: 8.15), NT: mean 51.75 (*SD*: 10.24), *p* = 0.064).

### 3.3. Analyses of Deviant Performance in the Four Domains

Please refer to Appendix A with an overview regarding the results of the two-sided Fisher’s exact tests, which we applied for the analysis as to whether the percentage of individuals with deviations differed significantly between the groups.

#### 3.3.1. Index Level

The index level revealed that more DYSL showed deviations in phonological, auditory, and cerebellar indices than NT (see Figure 1 and for another depiction of the individual performance within the groups see Appendix A). DYSL had significantly more individuals with deviations than NT in most phonological indices (*index phon* (DYSL: 90%/1 *SD* and 75%/1.65 *SD*; NT: 14%/1 *SD* and 0%/1.65 *SD*; 1 *SD*: *p* < 0.001; 1.65 *SD*: *p* < 0.001), *index RAN* (DYSL: 40%/1 *SD* and 25%/1.65 *SD*; NT: 0%/1 *SD* and 0%/1.65 *SD*; 1 *SD*: *p* = 0.001; 1.65 *SD*: *p* = 0.021). In the *index PWM*, DYSL had more individuals with deviations than NT for 1 *SD* only in a trend (DYSL: 35%/1 *SD* and 10%/1.65 *SD*; NT: 10%/1 *SD* and 0%/1.65 *SD*; 1 *SD*: *p* = 0.054; 1.65 *SD*: *p* = 0.232).

DYSL had significantly more individuals with deviations than NT in the *index auditory total* for 1 *SD* (DYSL: 20%/1 *SD* and 10%/1.65 *SD*; NT: 0%/1 *SD* and 0%/1.65 *SD*; 1 *SD*: *p* = 0.048; 1.65 *SD*: *p* = 0.232) and the *index auditory error* (DYSL: 85%/1 *SD* and 50%/1.65 *SD*; NT: 14%/1 *SD* and 10%/1.65 *SD*; 1 *SD*: *p* < 0.001; 1.65 *SD*: *p* = 0.005). 

In all indices addressing visual performances, 0% of the individuals of both groups showed deviations. In the *index cerebellar*, there were no significant differences regarding the number of individuals with deviations between the two groups (DYSL: 30%/1 *SD* and 15%/1.65 *SD*; NT: 10%/1 *SD* and 5%/1.65 *SD*; 1 *SD*: *p* = 0.130; 1.65 *SD*: *p* = 0.343).

#### 3.3.2. Subtest Level

However, when looking into the subtests of the different performance domains, there were deviant performances in almost all subtests in individuals from both groups. Whether or not significantly more DYSL than NT showed deviations at the subtest level differed strongly between the subtests (see Figure 2, Figure 3, Figure 4 and Figure 5).

In the case of phonological awareness, not all subtests showed a statistically significant difference with regard to the fact that more DYSL than NT showed deviations, as one would expect looking at the descriptive statistics (see Figure 2, (*pseudo segm* (DYSL: 100%/1 *SD* and 20%/1.65 *SD*; NT: 14%/1 *SD* and 5%/1.65 *SD*; 1 *SD*: *p* = 0.159; 1.65 *SD*: *p* = 0.184), *vow subst* (DYSL: 45%/1 *SD* and 25%/1.65 *SD*; NT: 19%/1 *SD* and 14%/1.65 *SD*; 1 *SD*: *p* = 1.000; 1.65 *SD*: *p* = 0.454) and *phon swap* (DYSL: 20%/1 *SD* and 15%/1.65 *SD*; NT: 14%/1 *SD* and 10%/1.65 *SD*; 1 *SD*: *p* = 0.697; 1.65 *SD*: *p* = 0.663)). For the other subtests of phonological awareness, differences in the percentage of individuals with deviations for phonological variables were significant (i.e., *sound cat* (DYSL: 75%/1 *SD* and 60%/1.65 *SD*; NT: 29%/1 *SD* and 10%/1.65 *SD*; 1 *SD*: *p* = 0.005; 1.65 *SD*: *p* < 0.001) or *3 syl PW rct* (DYSL: 80%/1 *SD* and 65%/1.65 *SD*; NT: 14%/1 *SD* and 5%/1.65 *SD*; 1 *SD*: *p* < 0.001; 1.65 *SD*: *p* < 0.001)). In *digit repetion*, which is a subtest for the assessment of phonological working memory, there was no significantly higher percentage of DYSL with deviations. Reaction times in RAN subtests showed a significantly higher percentage of DYSL with deviations than NT (i.e., *RAN letter rct* (DYSL: 65%/1 *SD* and 40%/1.65 *SD*; NT: 10%/1 *SD* and 5%/1.65 *SD*; 1 *SD*: *p* < 0.001; 1.65 *SD*: *p* < 0.001)), whereas errors in RAN subtests did not (i.e., *RAN letter error* (DYSL: 35%/1 *SD* and 10%/1.65 *SD*; NT: 24%/1 *SD* and 5%/1.65 *SD*; 1 *SD*: *p* = 0.505; 1.65 *SD*: *p* = 0.606)).

In auditory subtests, DYSL only showed significantly more individuals with deviations than NT in the errors made, not the reaction time (see Figure 3, *single tone error* (DYSL: 75%/1 *SD* and 55%/1.65 *SD*; NT: 14%/1 *SD* and 10%/1.65 *SD*; 1 *SD*: *p* < 0.001; 1.65 *SD*: *p* = 0.003), *tonse quence error* (DYSL: 75%/1 *SD* and 70%/1.65 *SD*; NT: 10%/1 *SD* and 5%/1.65 *SD*; 1 *SD*: *p* < 0.001; 1.65 *SD*: *p* < 0.001), *syllables error* (DYSL: 30%/1 *SD* and 30%/1.65 *SD*; NT: 14%/1 *SD* and 5%/1.65 *SD*; 1 *SD*: *p* = 0.277; 1.65 *SD*: *p* = 0.045)). This pattern of the higher percentage of DYSL with deviations than NT remained constant across the different ISIs of the tone sequences (see Appendix A).

For the visual magnocellular tests see Figure 4 and for the performance on the different speed levels see Appendix A). Only *CM rct* (DYSL: 5%/1 *SD* and 5%/1.65 *SD*; NT: 14%/1 *SD* and 10%/1.65 *SD*; 1 *SD*: *p* = 0.048; 1.65 *SD*: *p* = 0.488) showed significant differences in percentage of subjects with deviations between the groups, but here, NT had the higher percentage of individuals with deviances compared to DYSL. For *MG error*, although the DYSL showed deviations in 25% of subjects for 1 *SD*, 19% of NT also showed deviations for 1 *SD*, and thus no significant difference between the groups could be observed here.

Within the visual parvocellular tests, *PS error* was the only subtest in which DYSL had a significantly higher number of individuals with deviations (DYSL: 55%/1 *SD* and 40%/1.65 *SD*; NT: 10%/1 *SD* and 10%/1.65 *SD*; 1 *SD*: *p* = 0.003; 1.65 *SD*: *p* = 0.032).

For *tapping*, significantly more DYSL showed deviations at 1 *SD* but not for 1.65 *SD* (see Figure 5, DYSL: 45%/1 *SD* and 20%/1.65 *SD*; NT: 10%/1 *SD* and 10%/1.65 *SD*; 1 *SD*: *p* = 0.015; 1.65 *SD*: *p* = 0.410), whereas there was no significant difference for *statue* (DYSL: 30%/1 *SD* and 10%/1.65 *SD*; NT: 19%/1 *SD* and 5%/1.65 *SD*; 1 *SD*: *p* = 0.484; 1.65 *SD*: *p* = 0.481).

#### 3.3.3. Deviant Performance in the Individual Level Presented as Heatmaps

The “heatmaps” were introduced to examine and represent individual performance profiles (see Figure 6, Figure 7, Figure 8, Figure 9 and Figure 10 and hints how to read the heatmaps see Section 2.3.). For the depiction of the heatmaps showing individual profiles according to the 1.65 *SD* threshold, please refer to Appendix A.

As can be seen in Figure 6, 15 DYSL showed a combination of deviations in phonological as well as auditory performance, two DYSL showed auditory performance deviations only without phonological performance deviations, and three DYSL showed phonological performance deviations only without auditory performance deviations. Therefore, the predomination type of profile in DYSL regarding the 1 *SD* threshold was a combination of phonological and auditory performance deficits.

Looking at the performance in the phonological subtests (see Figure 7), there was one DYSL who had deviations in almost all phonological areas at the 1 *SD* threshold. All other DYSL showed variant profiles of deviations in the impaired phonological processing areas.

Regarding auditory performance (see Figure 8), on the 1 *SD* threshold, 17 DYSL individuals showed deviations in the range of the auditory errors. The processing auf auditory syllables was a very relevant task for the temporal auditory deficit theory. Six DYSL showed deviations in syllables and eleven DYSL showed no deviations in syllables but in tones and tone sequences.

For the visual performance (see Figure 9), 13 DYSL did not have a single magnocellular performance deviation. Fifteen DYSL also showed deviations in one subtest of the parvocellular domain.

Twelve DYSL showed deviations in cerebellar subtests. 

### 3.4. Predicting Reading Performance

Out of the variables used within the regression analysis to investigate which variables had the most predictive value for reading performance, only *index phon* (phonological awareness) and *index auditory error* showed significant predictive power (see Table 8). As described in Section 2.3, we inverted the phonological awareness parameter resulting in high values representing lower performance. When the phonological awareness score increased by one point (an increase means a worsening of the performance), reading performance decreased on average by 8.3 points (*β* = −8.3, 95% CI: −13.3 to −3.3, *p* < 0.001). When auditory errors increased by one point, reading performance decreased by 3.7 points (*β* = −3.7, 95% CI: −6.7 to −0.619, *p* = 0.018).

## 4. Discussion

The present study examined the performance of neurotypical normal reading and dyslexic children within a consistent orthography system (German) using a large-scale behavioral test battery operationalizing processing domains identified as critical by three major deficit theories (phonological, magnocellular, and cerebellar deficit theories). Four different kinds of analyses were conducted: (1) to reveal group differences in the various performance domains, (2) to compare the percentage of individuals with deviating performances between the two groups, (3) to depict performance profiles in the individuals, and (4) to define the parameter that was the best predictor of reading performance within the whole group. 

### 4.1. Which Performance Domain Showed the Most Impairments?

In the present sample, using the selected tests, there was clear evidence for impaired phonological processing as well as impaired auditory processing on the group as well as on the individual level in children with dyslexia. Impaired cerebellar performance can additionally be observed in some individuals, whereas there were only a few individuals showing visual deficits within those visual tasks that we selected. 

That DYSL showed deficits in phonological processing was very clearly visible at all levels of analysis. Most phonological performance deficits on the group level can be observed in tasks operationalizing phonological awareness. Phonological awareness is important for the reading acquisition across language systems [72], and deficits in this ability are highly associated with dyslexia. Melby-Lervåg et al. [73] demonstrated in a meta-analysis of 88 studies investigating phonological awareness in children with dyslexia that there was an overall significant mean effect of d = −1.37 in that children with dyslexia performed poorly on phonological awareness compared to children without reading difficulty. Additionally, studies examining different cognitive deficits in dyslexia in the same sample described similar findings of a dominant impairment of phonological processing in their respective samples [23,48,51].

Regarding auditory processing, the present study observed significant group differences for some of the auditory tasks. Two of the investigated auditory tasks addressed the processing capacity of short acoustic transient information: tone sequences with short ISIs and syllables (see [74]). There were significant group differences for syllables, and additionally a significantly higher percentage of DYSL showing deficient performance for syllables at the subtest level. Yet, DYSL made significantly more errors in all different lengths of the ISIs of tone sequences, not only for the short ones. Together, these findings do not point stringently towards a superordinate impairment of auditory temporal order processing capacities in the present sample. There is inconsistent evidence of studies demonstrating deficient auditory processing in children with dyslexia, with studies finding evidence for an auditory processing deficit in dyslexia [75], not at all [76], or only in subsamples [23,48,50], with the findings of the present study fitting in the latter category.

Impairments within the magnocellular visual system would be indicated by performance differences in tasks involving fast-moving and low-contrast stimuli, since these stimulus characteristics appeal to the visual magnocellular system. However, we only saw a significant group difference in a variable that appealed to the parvocellular system. Evidence for a visual magnocellular deficit is also inconsistent [16]. Gori et al. [30] demonstrated in their study that (1) motion perception is deficient in children with dyslexia, that (2) deficient motion perception predicts reading development, and (3) visual training improves reading skills.

Regarding cerebellar performance on the group level and the summary of performance on the index level, there were no significant performance differences between the groups. Analogous to the discussion of the previous deficit areas, the evidence base for a cerebellar deficit is also to be assessed as heterogeneous [31,77].

In sum, regarding the question on the major processing impairments observable in the present sample, the strongest empirical evidence can be found for phonological impairments accompanied by auditory impairments. However, due to the lower predictive power of auditory performance for the reading ability, auditory processing deficits did not appear to be superior to phonological processing deficits. 

### 4.2. Are There Typical Individual Performance Profiles?

Several multicausal studies depicted individual performance profiles as intersection diagrams in order to demonstrate interindividual variance of performance profiles [23,48,51]. In the present study, we chose the visualization of individual performance profiles in the way of “heatmaps,” which allowed an overview of deficient performance in the individual across all investigated subtests via coding deficient performance as 1 in a red cell and none-deficient performance with a 0 in a white cell.

Individual heatmaps demonstrated that there were individuals who showed deficits in all three areas of phonological processing, whereas for others, the focus of deficits was limited to phonological awareness. There were no dominant profiles visible for a so-called double deficit (simultaneous impairments of phonological awareness and RAN [78]). Regarding auditory performance, more DYSL showed worse performance for single tones and tone sequences than for syllables, a pattern not consistent with an overruling auditory deficit, as already discussed in Section 4.1. Regarding visual processing, there were individuals in both the NT and the DYSL with deficient performance in tasks that appeal to the magnocellular system, yet there was no dominating pattern of visual impairments regarding performance in the tests we used (in case visual impairments can be observed, these impairments were distributed across the different subtests). There was also no evidence that higher motion speed modulates these results (see Appendix A). Regarding cerebellar performances depicted in the individual heat maps, it can be observed that cerebellar performance deficits existed in at least one subtest (finger tapping) for a high percentage of DYSL. However, the percentage of individuals with deviant cerebellar performances did not differ significantly between the groups. According to Raberger and Wimmer [79], the association of dyslexia and balancing problems could be modulated by a comorbidity of ADHD. Since attention capacities are well controlled for in the present sample, this could be an explanation in that there were no differences in performance for the task *statue* in the present sample.

In sum, the present individual profiles did not reveal predominant deficit profiles in children with dyslexia that would allow for the classification of subgroups [46,47] and speak more for an individual profile of performance deficits in children with dyslexia, as also described by others [18].

### 4.3. Which Domains Were Predictors for Reading Performance?

The strongest predictor for reading ability over the whole sample was phonological awareness, followed by auditory processing. That phonological skills were the strongest predictor for reading ability is in line with the findings of other studies. Melby-Lervåg et al. [73] demonstrated in a meta-analysis the important role of phonological awareness for differences in reading acquisition. The study of Carroll et al. [50] used a longitudinal study design that allows for the conclusion of a causal relationship [16]. Carroll et al. [50] observed that print knowledge, verbal short-term memory, phonological awareness, and rapid naming were good predictors of later poor reading. Debska et al. [48] also described that phonological awareness and RAN were related to reading abilities, whereas the other investigated skills did not explain additional variance.

On the subtest and individual levels, cerebellar as well as visual deficits were observed, but statistically they had no predictive value for the reading performance, which is contrary to the findings of Gori et al. [30]. They observed that the capacity of the prereading motion perception predicted future reading development.

### 4.4. Methodological Issues

There were several methodological issues that limited the comparability of the present results to findings of previous research. Previous studies used different tasks and paradigms, and thereby they differed in the way that they measure aspects of cognitive or sensoric processes [23,47,48,50,51]. Due to ethical considerations regarding the amount of work children can be expected to do in a research study, we made a selection of tasks for our test battery. Thus, we do not claim to have covered all processing areas that are discussed as critical in the context of dyslexia.

Additionally, findings on performance differences strongly depended on the degree of resolution of the data and the statistical approach with which these performances were investigated (group means vs. individual profiles, aggregation of performance in a domain in an index vs. subtest level). The more aggregated the parameter (i.e., sum scores comprised of different subtests the same domain), the more one runs the risk that individual deficits in specific processing areas will not be revealed.

Some previous studies computed indices (i.e., z-standardized mean scores for a specific performance domain) in order to allow the comparison of performances between different domains and the classification of performance deficits [23,50]. However, these studies differed in the way these indices were computed. Some studies standardized the data using the mean and *SD* of the total sample [50], whereas others used those of the control sample or even applied a three-step-procedure [23] (first step: standardization with mean and *SD* of the total sample, second step: eliminate those controls with deviating performance, third step: renewed standardization using the mean and *SD* of the adjusted control sample).

Finally, previous studies differed in the threshold used to label performances as deviant: while some used 1.65 *SD* [23], others used 1 *SD* [50], arguing that most standardized tests take 1 *SD* below average as salient. For better comparison, we depicted in the present study both thresholds (1 *SD* and 1.65 *SD*). As evident in our figures, this differentiation quite strongly influenced the extent to which deficient performance was observed. As can be seen in Figure 1, the deviations in the index phon and index auditory error were observed at 1 *SD* for almost the same number of DYSL (90%/index phon and 85%/index auditory error), but at the harder deviation criterion of 1.65 *SD*, the number of subjects in the DYSL decreased more in the index auditory error (to 50%) than in index phon (75%).

## 5. Conclusions

The present study demonstrated that phonological processing as well as auditory processing capacities have a major influence on reading ability in the present sample of children within a consistent orthography. Additionally, the data of the present study confirm phonological and auditory processing impairments for children with dyslexia on group, index, subtest, and individual levels. It can thus be concluded that phonological deficits are associated with dyslexia in a consistent orthography system accompanied by auditory deficits. Besides these prominent phonological and auditory processing impairments, we also acknowledge that the individual profiles depicted in the present heatmaps also point out the individual heterogeneity of the disorder, which has also been acknowledged by others studies [18,23,30,50].

Even though the design of the present study does not allow to speak of “causes” for the observed processing deficits in dyslexia, and it also cannot be claimed that all paradigms that are currently discussed about dyslexia have been investigated, the fact that observed deficits that are still present in dyslexic children with several years of reading experience demonstrates the importance of these domains for the reading and spelling ability.

Future research should investigate individual deficit profiles longitudinally, with studies starting before literacy acquisition, at as many processing domains as possible. These individual deficit profiles should then be used to select appropriate interventions to promote reading and spelling.

## Figures and Tables

**Figure 1 brainsci-12-01292-f001:**
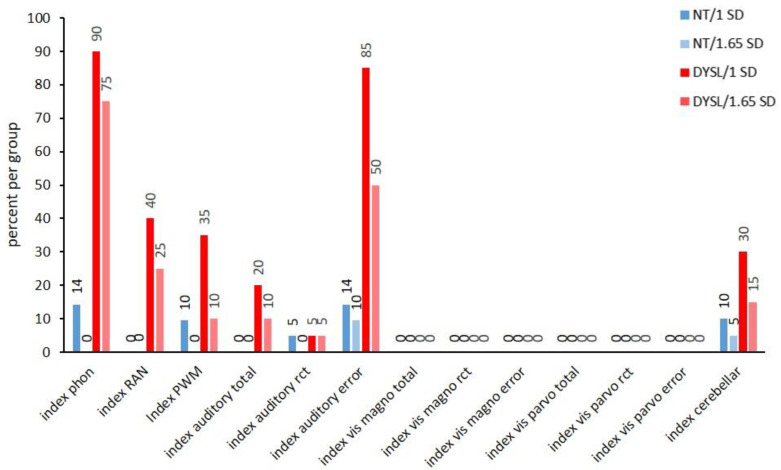
Deviant performance in percent per group at the index level. NT: neurotypical readers, DYSL: dyslexic readers, number above bars: percentage of individuals with deviating performance in the respective index, phon: phonological awareness, RAN: rapid automatized naming, PWM: phonological working memory, rct: reaction time, magno: magnocellular, parvo: parvocellular.

**Figure 2 brainsci-12-01292-f002:**
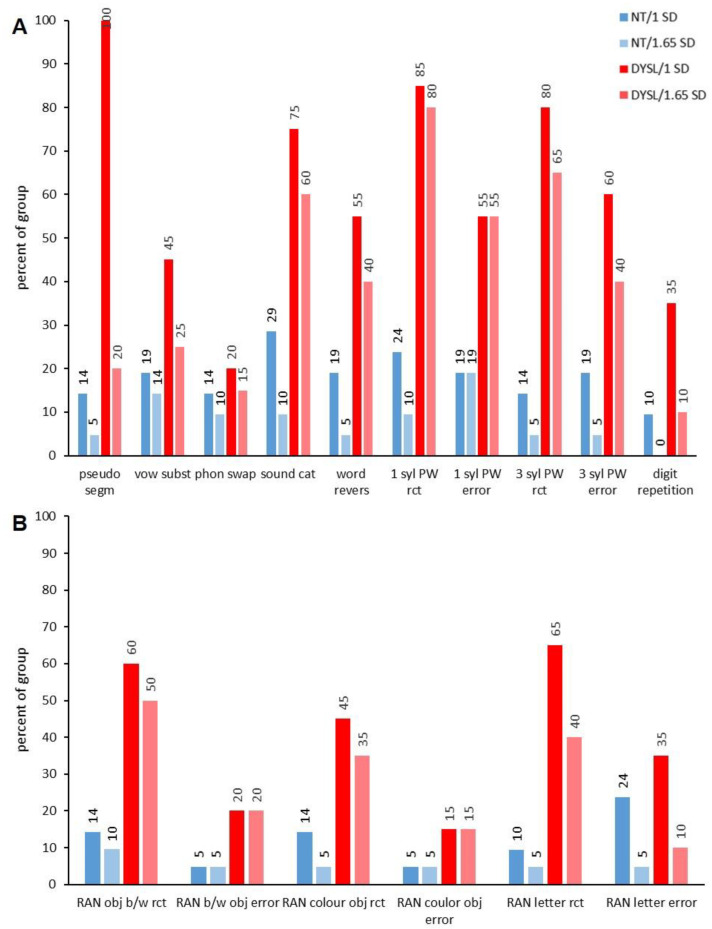
Deviating performance/phonological subtests. (**A**) Subtests of phonological awareness and phonological working memory. (**B**) Subtests of rapid automatized naming. NT: neurotypical normal reading children, DYSL: children with dyslexia, pseudo segm: segmentation of pseudowords, vow subst: vowel substitution, phon swap: swapping of phonemes, sound cat: judging similarity of speech sound categories, word revers: reverse speech sounds of a word, syl: syllables, PW: pseudowords, RAN: rapid naming, b/w: black and white, obj: object, rct: reaction time.

**Figure 3 brainsci-12-01292-f003:**
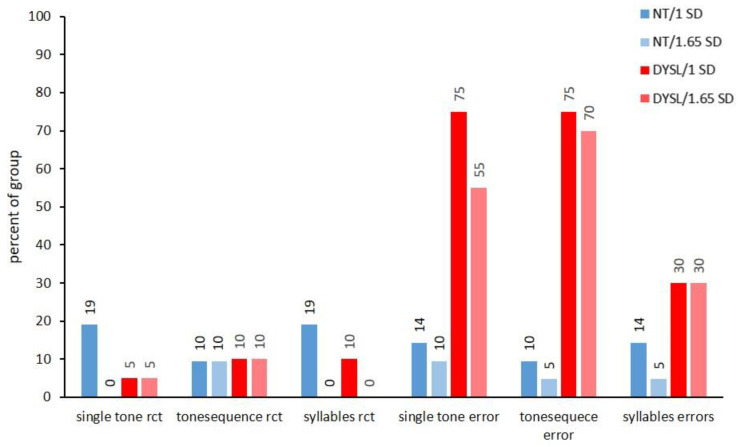
Deviating performance/auditory subtests. Auditory subtests. NT: neurotypical normal reading children, DYSL: children with dyslexia, rct: reaction time.

**Figure 4 brainsci-12-01292-f004:**
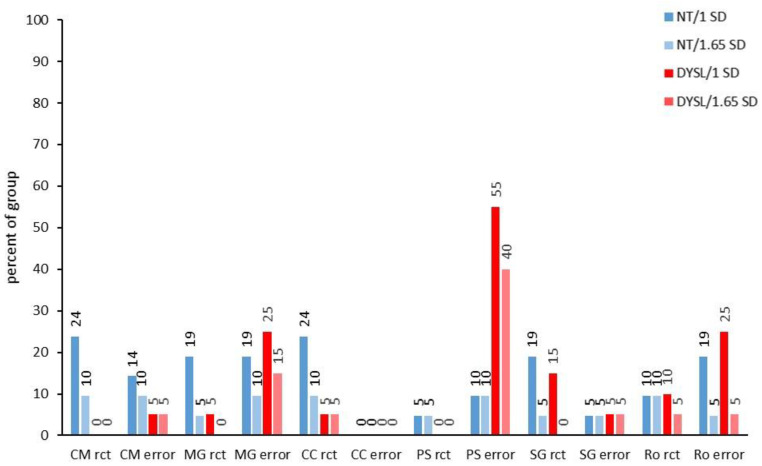
Deviating performance/visual subtests. NT: neurotypical normal reading children, DYSL: children with dyslexia, rct: reaction time, err: errors, MG: moving gratings, CM: coherent motion, CC: coherent color, PS: pattern sequence, StG: stationary gratings, Ro: mental rotation.

**Figure 5 brainsci-12-01292-f005:**
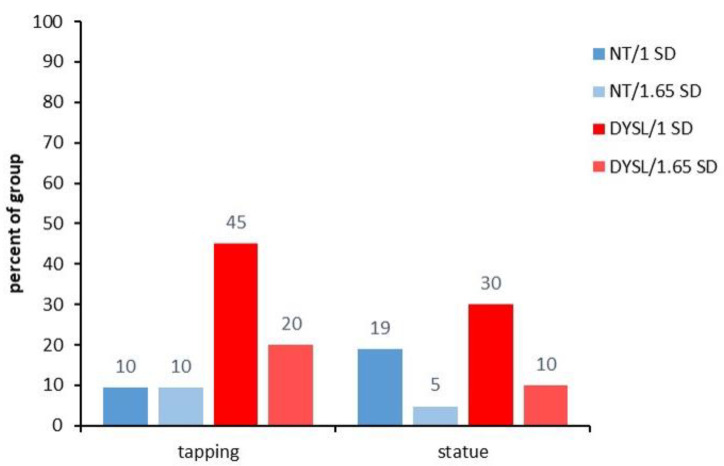
Deviating performance/cerebellar subtests. NT: neurotypical normal reading children, DYSL: children with dyslexia.

**Figure 6 brainsci-12-01292-f006:**
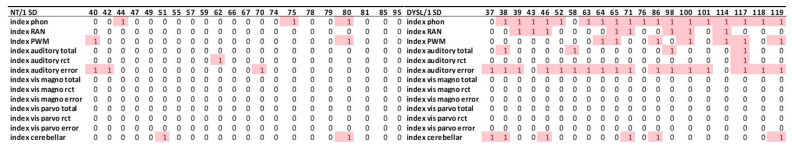
Deviating individual performance/indices. NT: neurotypical normal reading children, DYSL: children with dyslexia (numbers in the head row depict the individual subject), phon: phonological awareness, RAN: rapid automatized naming, PWM: phonological working memory, rct: reaction time, magno: magnocellular, parvo: parvocellular.

**Figure 7 brainsci-12-01292-f007:**
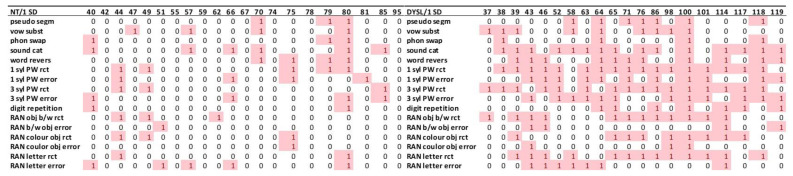
Deviating individual performance/phonology/subtest level. NT: neurotypical normal reading children, DYSL: children with dyslexia (numbers in the head row depict the individual subject), rct: reaction time, err: errors, Phon Mean: mean of the phonological subtests for phonological processing in an inner sense, syl: syllables, PW: pseudowords, RAN: rapid naming, b/w: black and white, obj: object.

**Figure 8 brainsci-12-01292-f008:**
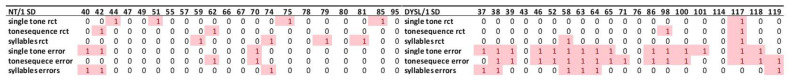
Deviating individual performance/auditory/subtest level. NT: neurotypical normal reading children, DYSL: children with dyslexia (numbers in the head row depict the individual subject), rct: reaction time, err: errors, TS: tone sequence ISI: interstimulus interval.

**Figure 9 brainsci-12-01292-f009:**
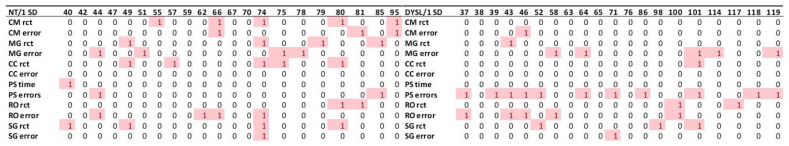
Deviating individual performance/visual/subtest level. NT: neurotypical normal reading children, DYSL: children with dyslexia (numbers in the head row depict the individual subject), rct: reaction time, err: errors, CM: coherent motion, MG: moving gratings, CC: coherent color, PS: pattern sequence, SG: stationary gratings, Ro: rotation.

**Figure 10 brainsci-12-01292-f010:**
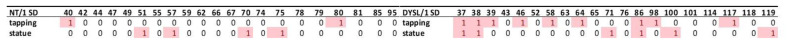
Deviating individual performance/cerebellar/subtest level. NT: neurotypical normal reading children, DYSL: children with dyslexia (numbers in the head row depict the individual subject).

**Table 1 brainsci-12-01292-t001:** Overview of the three theories in dyslexia investigated.

Deficit Theory	References	Presumed Neurobiological Background	Explanation for Dyslexia-Specific Performance Problems	Behavioral Deficits in Readers with Dyslexia
**Phonological deficit**	[28] Snowling, M. J. (1998)	left-hemisphere perisylvian areas of the reading network	deficient acquisition of grapheme–phoneme conversion, reading, and spelling deficits	poor phonological performance (phonological awareness skills, phonological working memory, access to semantic memory)
**Magnocellular deficit—visual**	[37] Eden et al. (1996)	visual magnocellular system	unstable eye movements hinder word perception, unstable visual perception impairs transformation of visual letter to speech sound—phonological deficits are a secondary consequence	worse performance of fast-moving, low-contrast visual stimuli, unstable visual perception, unstable eye control
**Magnocellular deficit—auditory**	[40] Tallal, P. (1980) [41] Stein and Talcott (1999)	auditory (magnocellular) system	deficient frequency discrimination hinders disassembly of the auditory speech—phonological deficits are a secondary consequence	deficient ability to process fast-changing acoustic stimuli
**Cerebellar deficit**	[42] Nicolson and Fawcett (1990)	cerebellum	impairment in automation of learned skills—phonological deficits are the result of impairment in the automation of grapheme–phoneme mapping	motoric clumsiness, worse performance in tasks requiring the automation of motor processes

**Table 2 brainsci-12-01292-t002:** Sample criteria.

	NT Mean (*SD*)	DYSL Mean (*SD*)	Statistics	*p*-Value
*n*	21	20		
Age	11.48 (0.75)	11.38 (1.08)	t (33.65) = 0.33	*p* = 1.000
Nonverbal IQ	115.86 (12.18)	114.40 (9.81)	t (37.98) = 0.42	*p* = 1.000
Spelling performance	107.62 (11.41)	71.85 (12.10)	t (38.55) = 9.73	*p* < 0.01
Reading performance	106.49 (5.00)	78.71 (7.45)	t (33.03) = 13.95	*p* < 0.01
Discrepancy IQ and spelling	0.55 (0.80)	2.84 (1.04)	t (35.62) = -7.85	*p* < 0.01
Discrepancy IQ and reading	0.63 (0.78)	2.38 (0.64)	t (38.24) = -7.90	*p* < 0.01
Test score handedness	47.00 (19.17)	43.72 (14.15)	t (36.76) = 0.62	*p* = 1.000

NT: neurotypical normal reading children, DYSL: children with dyslexia, IQ: intelligence quotient; test score of spelling and reading performance is transferred on IQ norm scale via z-transformation, discrepancy in standard deviation.

**Table 3 brainsci-12-01292-t003:** Attention performance.

	NT	DYSL	Statistics	*p*-Value
	Mean (*SD*)	Mean (*SD*)		
*n*	21	20		
Distractability rct	536.48 (64.37)	530.30 (74.94)	t (37.50) = 0.28	*p* = 1.000
Sustained attention rct	688.24 (160.51)	665.75 (93.70)	t (32.49) = 0.55	*p* = 1.000
Gonogo rct	475.67 (77.34)	457.80 (53.96)	t (35.82) = 0.86	*p* = 1.000
Distractability cor	39.19 (1.60)	39.40 (0.82)	t (30.15) = −0.53	*p* = 1.000
Distractability com	4.14 (3.81)	7.05 (7.31)	t (28.31) = −1.59	*p* = 1.000
Distractability om	0.81 (1.60)	0.60 (0.82)	t (30.15) = 0.53	*p* = 1.000
Sustained attention cor	47.71 (2.67)	46.75 (3.46)	t (35.71) = 1.00	*p* = 1.000
Sustained attention com	2.24 (3.42)	4.90 (6.51)	t (28.42) = −1.63	*p* = 1.000
Sustained attention om	2.29 (2.67)	3.25 (3.46)	t (35.71) = −1.00	*p* = 1.000
GoNogo cor	19.95 (0.22)	19.80 (0.52)	t (25.16) = 1.21	*p* = 1.000
GoNogo om	0.95 (0.92)	1.60 (1.60)	t (30.00) = −1.58	*p* = 1.000
GoNogo com	0.05 (0.22)	0.20 (0.52)	t (25.16) = −1.21	*p* = 1.000

NT: neurotypical normal reading children, DYSL: children with dyslexia, rct: reaction time, cor: correct response, om: omission errors, com: commission errors.

**Table 4 brainsci-12-01292-t004:** Phonological performance.

	NT	DYSL	Statistics	*p*-Value
	Mean (*SD*)	Mean (*SD*)		
*n*	21	20		
Phonological Awareness				
Pseudo Segm	7.57 (1.86)	6.23 (2.79)	t (30.91) = 1.79	*p* = 1.000
Vow Subst	8.57 (1.36)	7.01 (2.16)	t (29.82) = 2.79	*p* = 0.085
Phon Swap	7.71 (1.95)	6.26 (2.15)	t (36.57) = 2.28	*p* = 0.914
Sound Cat	9.10 (1.04)	6.90 (2.13)	t (25.62) = 4.20	*p* <0.001
Word Revers	6.76 (2.45)	3.53 (2.93)	t (35.24) = 3.85	*p* < 0.001
1 syl PW rct	11.10 (3.53)	22.75 (8.66)	t (24.91) = −5.59	*p* < 0.001
1 syl PW error	1.00 (1.14)	2.65 (1.87)	t (31.11) = −3.39	*p* < 0.017
3 syl PW rct	31.14 (11.21)	55.20 (19.82)	t (29.73) = −4.75	*p* < 0.001
3 syl PW error	2.81 (2.82)	8.60 (11.58)	t (21.15) = −2.18	*p* = 0.510
Phonological recoding in working memory				
digit repetition	12.60 (3.63)	9.20 (2.09)	t (30.37) = 3.72	*p* < 0.001
Phonological recoding with access to the semantic lexicon				
RAN obj b/w rct	22.33 (3.12)	28.80 (7.87)	t (24.59) = −3.43	*p* < 0.017
RAN b/w obj error	0.05 (0.22)	0.25 (0.55)	t (24.60) = −1.53	*p* = 1.000
RAN colour obj rct	26.33 (5.56)	33.60 (11.62)	t (26.98) = −2.53	*p* = 0.187
RAN colour obj error	0.05 (0.22)	0.25 (0.72)	t (22.34) = −1.21	*p* = 1.000
RAN letter rct	25.67 (5.28)	34.50 (13.72)	t (24.27) = −2.70	*p* = 0.119
RAN letter error	0.62 (0.97)	1.30 (1.42)	t (33.48) = −1.78	*p* = 1.000

NT: neurotypical normal reading children, DYSL: children with dyslexia, RAN: rapid naming, b/w: black and white, obj: object, rct: reaction time, PW: pseudowords, syl: syllables, Pseudo Segm: segmentation of pseudowords, Vow Subst: vowel substitution, Phon Swap: swapping of phonemes, Sound Cat: judging similarity of speech sound categories, Word Revers: reverse speech sounds of a word.

**Table 5 brainsci-12-01292-t005:** Auditory performance.

	NT	DYSL	Statistics	*p*-Value
	Mean (*SD*)	Mean (*SD*)		
*n*	21	20		
Single tone rct	570.79 (72.38)	481.09 (97.02)	t (35.11) = 3.34	*p* < 0.01
Tone sequence rct	1043.84 (330.72)	1050.56 (393.38)	t (37.18) = −0.06	*p* = 1.000
Syllable rct	607.54 (83.11)	538.63 (130.97)	t (31.91) = 2.00	*p* = 0.270
Single tone error	6.67 (5.97)	16.35 (7.82)	t (35.55) = −4.44	*p* < 0.001
Tone sequence error	11.95 (15.23)	46.60 (25.27)	t (30.90) = −5.29	*p* < 0.001
Syllable errors	3.76 (4.76)	9.20 (7.14)	t (32.90) = −2.85	*p* < 0.001

NT: neurotypical normal reading children, DYSL: children with dyslexia, rct: reaction time.

**Table 6 brainsci-12-01292-t006:** Visual performance.

	NT	Dysl	Statistics	*p*-Value
	Mean (*SD*)	Mean (*SD*)		
*n*	21	20		
Magnocellular System				
CM rct	742.78 (135.60)	709.09 (99.37)	t (36.50) = 0.92	*p* = 1.000
CM error	19.76 (28.53)	24.249 (24.28)	t (37.87) = −0.55	*p* = 1.000
MG rct	717.49 (125.29)	721.99 (77.87)	t (33.68) = −0.14	*p* = 1.000
MG error	23.24 (2.79)	24.659 (4.64)	t (30.88) = −1.17	*p* = 1.000
Parvocellular System				
CC rct	571.51 (55.29)	577.19 (54.95)	t (37.65) = −0.33	*p* = 1.000
CC error	0.00 (0.00)	0.09 (0.23)	t (18.00) = −1.20	*p* = 1.000
PS rct	739.29 (536.87)	668.99 (240.96)	t (28.34) = 0.55	*p* = 1.000
PS error	2.00 (2.12)	5.39 (3.83)	t (27.46) = −3.46	*p* < 0.006
StG rct	622.29 (126.91)	622.29 (95.46)	t (37.06) = 0.00	*p* = 1.000
StG error	0.86 (3.05)	0.59 (1.47)	t (29.08) = 0.41	*p* = 1.000
Other Visual Control Tasks				
Ro rct	1464.30 (491.18)	1404.9 (409.74)	t (38.35) = 0.42	*p* = 1.000
Ro error	11.71 (5.17)	13.709 (4.16)	t (37.96) = −1.36	*p* = 1.000

NT: neurotypical normal reading children, DYSL: children with dyslexia, rct: reaction time, CM: coherent motion, CC: coherent color, PS: pattern sequence, MG: moving gratings, StG: stationary gratings, Ro: mental rotation.

**Table 7 brainsci-12-01292-t007:** Cerebellar performance.

	NT	DYSL	Statistics	*p*-Value
	Mean (*SD*)	Mean (*SD*)		
*n*	21	20		
Tapping	45.57 (8.15)	51.75 (10.24)	t (36.28) = −2.13	*p* = 0.064
Statue	29.24 (1.61)	28.55 (1.85)	t (37.68) = 1.27	*p* = 0.406

NT: neurotypical normal reading children, DYSL: children with dyslexia.

**Table 8 brainsci-12-01292-t008:** The prediction of reading performance by a multiple linear regression analysis over the whole sample revealed two significant predictors: phonological awareness and auditory errors.

	Regression Coefficient B	Standard Error	*t*	*p*	95.0% Confidence Interval for B	
					Upper	Lower
Index phon	−8.285	2.513	−3.298	<0.001	−13.210	−3.361
Index PWM	−0.748	2.262	−0.331	0.741	−5.181	3.684
Index RAN	−2.026	1.466	−1.382	0.167	−4.900	0.848
Index auditory rct	0.348	2.012	0.173	0.863	−3.596	4.293
Index auditory error	−3.669	1.556	−2.357	0.018	−6.719	−0.619
Index vis magno total	−2.521	2.846	−0.886	0.376	−8.100	3.058
Index vis parvo total	−0.857	2.500	−0.343	0.732	−5.756	4.042
Index cerebellar	2.735	1.825	1.498	0.134	−0.843	6.312

phon: phonological awareness, PWM: phonological working memory, RAN: rapid automatized naming, rct: reaction time, vis: visual, magno: magnocellular, parvo: parvocellular.

## Data Availability

The data presented in this study are available on request from the corresponding author. The data are not publicly available due to local regulations on data protection regarding children’s medical data.

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
