# Peer review of "Multiple Case Studies in German Children with Dyslexia: Characterization of Phonological, Auditory, Visual, and Cerebellar Processing on the Group and Individual Levels"

_brainsci, 2022, doi:10.3390/brainsci12101292_

Round 1
Reviewer 1 Report
The article is devoted to the study of phonological, auditory, visual, and cerebellar processing in children with dyslexia. The article contains valuable scientific material. I have some important comments for paper revisions.
Authors should remove too frequent "our" from the abstract. References should be added to the article in the last paragraph of the first page (lines 37-43).
To systematize the material in the Introduction, the authors should add a comparative table on the hypotheses of dyslexia.
Lines 153-154, 333, 338-339 - there are errors in links.
Tables 1 and 2, 5, 6: it is not clear why the p values ​​are like that (<0.01 and > 0.99, this is very strange).
I advise the authors to give a meaningful title to paragraph 3.3.3. Figures 6-10 should also be more specific and meaningful. It is difficult for the reader to understand their essence.
There are very few references in the Discussion. Authors should compare their results with different studies.
Author Response
Dear Reviewer,
thank you very much for your time and helpful comments. We believe that due to the revisions the manuscript improved significantly.
Please find the detailed description of the changes in the manuscript in the attached letter.
Sincerly
Carolin Ligges

Reviewer 2 Report
Dear Authors,
I read your work entitled "Multiple case study in German children with dyslexia: characterisation of phonological, auditory, visual, and cerebellar processing on the group and individual level" and here i enclose my recommendations to you:
1. The introduction is well written but there is a need for more references since there are several sentences without literature support.
2. In Methods Section paragraph 1 I suggest the Authors to place it in the Results section. The section 2.2 is too long I suggest the Authors to make it more core and sure there is space for improvement. I also suggest to have a separate paragraph (e.g., 2.3.) of the statistic analysis that was followed in this study.
3. In Results section report the Non-significant p-levels as NS or the exact number using superscripts. I also suggest some of the Figures to be entered in a Appendix.
4. The Discussion is well written but again the same issues as Introduction need more literature support and the Discussion must be better connected with the Introduction.
Thank you.
Author Response

(The authors gave the same response as above.)

Round 2
Reviewer 1 Report
Dear authors,
Thank's for article revision according to my comments.
I only advice you to add references into Table 1.
Author Response
Dear reviewer, please find attached the new version of the manuscript. I have added the references I quoted in the manuscript regarding the deficit theories to the table. For better readability, I have included the references both in numerical citation style and with the authors' names.
Thank you again for your advice,
sincerely
Carolin Ligges
